# Design and Evaluation of An Extended-Release Olmesartan Tablet Using Chitosan/Cyclodextrin Composites

**DOI:** 10.3390/pharmaceutics11020082

**Published:** 2019-02-15

**Authors:** Makoto Anraku, Ryo Tabuchi, Miwa Goto, Daisuke Iohara, Yasuyuki Mizukai, Yuji Maezaki, Akihiro Michihara, Daisuke Kadowaki, Masaki Otagiri, Fumitoshi Hirayama

**Affiliations:** 1Faculty of Pharmaceutical Sciences, Sojo University, 4-22-1 Ikeda, Nishi-ku, Kumamoto 860-0082, Japan; sojo_tabuchi@yahoo.co.jp (R.T.); chaiahchai@gmail.com (M.G.); dio@ph.sojo-u.ac.jp (D.I.); d-kado@ph.sojo-u.ac.jp (D.K.); otagirim@ph.sojo-u.ac.jp (M.O.); fhira@ph.sojo-u.ac.jp (F.H.); 2DDS Research Institute, Sojo University, 4-22-1 Ikeda, Nishi-ku, Kumamoto 860-0082, Japan; 3Nippon Kayaku Food Techno Co., Ltd., 219, Iwahanamachi, Takasaki, Gunma 370-1208, Japan; yasuyuki.mizukai@nipponkayaku.co.jp (Y.M.); yuji.maezaki@nipponkayaku.co.jp (Y.M.); 4Faculty of Pharmacy and Pharmaceutical Sciences, Fukuyama University, 1 Sanzo, Gakuen-cho, Fukuyama 729-0292, Japan; mitihara@fupharm.fukuyama-u.ac.jp

**Keywords:** chitosan, sulfobutyl ether β-cyclodextrin, extend-release, matrix tablet, antihypertensive effects

## Abstract

Sustained-release olmesartan tablets (OLM) were prepared by the simple, direct compression of composites of anionic sulfobutyl ether-β-cyclodextrin (SBE-β-CD) and cationic spray-dried chitosan (SD-CS), and were evaluated for use as a sustained release preparation for the treatment of hypertension. An investigation of the interaction between OLM and SBE-β-CD by the solubility method indicated that the phase diagram of the OLM/SBE-β-CD system was the A_L_ type, indicating the formation of a 1:1 inclusion complex. The release of OLM from tablets composed of the SD-CS/SBE-β-CD composite was slow in media at both pH 1.2 and at 6.8. The *in vitro* slow release characteristics of the SD-CS/SBE-β-CD composite were reflected in the *in vivo* absorption of the drug after normal rats were given an oral administration of the preparation. Furthermore, the SD-CS/SBE-β-CD composite continuously increased the antihypertensive effect of OLM in hypertensive rats, compared with that of the drug itself. These results suggest that a simple mixing of SD-CS and SBE-β-CD can be potentially useful for the controlled release of a drug for the continuous treatments of hypertension.

## 1. Introduction

Olmesartan (OLM), an angiotensin (Ang) II type 1 receptor antagonist, independently inhibits not only hypertension, but also albuminuria and glomerular hypertrophy, and functions independently of blood pressure [1,2]. Furthermore, Kadowaki et al. reported on the antioxidant activity of OLM *in vivo*, but this activity had direct and indirect antioxidant activity, including modulation via NADPH oxidase activity [3,4,5]. Thus, OLM has recently attracted attention as an effective antihypertensive drug with multifaceted effects, as reported above. Therefore, a more efficient design of OLM using new excipients would lead to the more effective and safe administration of the drug.

Chitosan (CS), a naturally occurring polysaccharide that is largely obtained from marine crustaceans, is a promising extended-release excipient for therapeutics and diagnostics, owing to its biocompatibility, biodegradability, low toxicity, and structural variability [6,7]. A new type of polysaccharide nano-carrier consisting of the polysaccharide CS and cyclic oligosaccharides such as cyclodextrins (CDs) was recently reported [8,9,10]. For example, CS nanoparticles containing an anionic cyclodextrin, sulfobutyl ether-β-cyclodextrin (SBE-β-CD), appear to have some potential for use as peptide carriers, because they combine improved peptide loadings with the capacity to promote peptide transport through the intestine, as observed in a frog intestinal sac model [11,12,13]. It is known that SBE-β-CD forms inclusion complexes with various drug molecules and can be used to solubilize drugs that are poorly water-soluble, thus making them more soluble than the parent compound, β-CD, due to the presence of a hydrophilic sulfobutyl moiety [14]. It therefore appears that CS/SBE-β-CD nano-carriers would be useful for preparing homogeneous nano-carriers that contain drugs that are poorly water soluble. On the other hand, one drawback to this system is that these nano-carriers are difficult to prepare, because they are too small in size to permit their isolation by simple filtration through filter paper. Furthermore, the extent of encapsulation of medicines in nano-carriers is not always high. In recent studies, we reported that a simple blend of CS and SBE-β-CD retarded the release of famotidine, a histamine H_2_ receptor antagonists, from ordinary tablets and the slow release of the drug was clearly reflected in *in vivo* absorption after oral administration to rats [15]. However, there are only a few reports concerning relationships between *in vivo* pharmacokinetics and *in vivo* pharmacodynamics using CS/SBE-β-CD composites.

In this study, sustained-release tablets containing OLM were prepared by direct compression, using composites of cationic CS and anionic SBE-β-CD, and the resulting preparation was then evaluated for use as a sustained release tablet for the treatment of hypertension using rats as model animals.

## 2. Materials and Methods

### 2.1. Materials 

CS (Molecular weight: 30 kDa) and SBE-β-CD (Captisol^®^, degree of substitution (DS) 7) was obtained from Nippon Kayaku Food Techno Co., Ltd. (Takasaki, Japan) and New Product Development Ligand Pharmaceuticals Incorporated (Lawrence, KS, USA), respectively. OLM was obtained from Nipro Co., Ltd. (Osaka, Japan). 2-Hydroxypropyl-β-CD (HP-β-CD, with a degree of substitution (D.S.) of the 2-hydroxypropyl group of 5.6) was a gift from Nihon Shokuhin Kako Co., Ltd. (Shizuoka, Japan). Spray-dried chitosan (SD-CS) was used in all experiments. All other chemicals were reagent grade or better.

### 2.2. Measurements of Turbidity

The CDs/SD-CS ratio in the complex was examined by monitoring the transmittance of the solution at a wavelength of 600 nm using a spectrophotometer (UV-1601 spectrophotometer, Shimadzu, Kyoto, Japan). Aqueous 1% acetic acid solutions of SBE-β-CD or HP-β-CD and SD-CS were mixed at different weight ratios. Each mixture was shaken vigorously. The mixtures were then left to stand for 10 min before measuring the transmittance, as a function of various mixing ratios.

### 2.3. Solubility Studies

Solubility measurements were conducted following a method of Higuchi and Connors [16,17]. Excess amounts (15 mg) of OLM were added to different concentrations of CD solutions (1.0 mL) and the suspension was shaken at 25 °C for 2 days. After reaching equilibrium, the vials were centrifuged for 5 min at 1000 *g* and the supernatant was filtered through a 0.2 μm-filter. The resulting filtrate was appropriately diluted and the level of OLM determined at a wavelength of 248 nm by means of a UV spectrometer (Shimadzu Scientific Instrument, Kyoto, Japan). The stability constant (Kc) of the CD complexes was calculated using the equation of Kc = slope/[intercept (1−slope)] using a slope and an intercept of the initial straight-line portion of the phase solubility diagrams [18].

### 2.4. Fourier Transform Infrared (FT-IR) Spectroscopy Study

The infrared absorption spectra of SBE-β-CD, SD-CS and their composites were obtained with a Fourier transform infrared spectroscopy (FT-IR) spectrophotometer (LX30-7012, Perkin Elmer, Waltham, MA, USA). Pellets were prepared by pressing the sample with potassium bromide.

### 2.5. Preparation of SD-CS/CDs/OLM Extended-Release Tablets

CS was dissolved in aqueous acetic acid (1%) resulting in the formation of the protonated species, and the resulting solutions were spray-dried using a SD-1000 instrument (Tokyo Rikakikai Co., Ltd, Tokyo, Japan) under the following conditions: inlet temperature of 140 °C, drying air flow of 0.50 m^3^/min, atomizing air pressure of 50 kPa, and an outlet temperature of 90–95 °C [19]. The extended-release SD-CS/ CDs tablets, with a total weight of 95 mg, were prepared using a mixture of OLM (5 mg) and an excipient (90 mg) at a ratio of 1:19. The mixture was compressed using a hydraulic press with a 7 mm diameter and compressing the mixed composite using a hydraulic press with a 7 mm diameter and a 2.0 mm thickness. The compression force was 10 kN/cm^2^ with a dwell time of 5 min. CDs, SD-CS, lactose, and three types of SD-CS/CDs composites were used as excipients.

### 2.6. In Vitro Dissolution Studies of SD-CS/CDs/OLM Composite

A dissolution test was carried out using a dissolution tester (NTR-6600, Toyama, Inc., Osaka, Japan). The rate of OLM dissolution was measured using the USP paddle method at 50 rpm using 450 mL of a pH 1.2 or pH 6.8 medium at 37 °C. An aliquot (1.0 mL) was automatically withdrawn, diluted appropriately with water and the level of OLM determined at a wavelength of 248 nm using a UV spectrometer (Shimadzu Scientific Instrument, Kyoto, Japan).

### 2.7. In Vivo Pharmacokinetic Studies of SD-CS/CDs/OLM Composite

ALL animal experiments, including 2.8, were performed according to the guidelines for the care and use of experimental animals under the approval of the Animal Research Committee of Sojo University (Permission No.: 2017-P-026). Furthermore, the study protocol also complied with the laws and notifications of the Japanese government prior to the commencement of the study. The rats (SD rats, *N* = 16, male, weight 250–350 g) were divided into four groups and fasted overnight. Each sample was immediately administered at a dose of 8 mg/kg/1 mL as suspensions containing SD-CS/CDs /OLM composite by oral gavage. Blood samples were collected from the tail vein for a period of up to 24 h. The plasma samples were prepared for HPLC analysis as reported previously, with minor modifications [20]. The pharmacokinetics and statistical analyses were computed by fitting using the Practical Pharmacokinetic Program (MULTI, a normal least square program [21]). 

### 2.8. Antihypertensive Studies of Spray-dried Chitosan Cyclodextrins (SD-CS/CDs)/Olmesartan Composite

20-week-old male stroke-prone spontaneously hypertensive (SHRsp)/Izm rats were used in the experiments (Japan SLC, Shizuoka, Japan). The rats (SHRsp/Izm rats, *N* = 20, male, weight 250–350 g) were divided into five groups and fasted overnight. The rats were treated with OLM 1 mg/kg/1 mL as suspensions in each of the composites (*n* = 4). The water administration group was used as a control (*N* = 4). Mean blood pressure (MBP) was measured for 24 h in conscious rats by the indirect tailcuff method (BP-98A; Softron, Tokyo, Japan) without anesthesia.

### 2.9. Statistics

Results are reported as the mean ± SEM. Statistical significance was evaluated using analysis of variance (ANOVA), followed by the Tukey-Kramer post hoc test. For all analyses, *P* < 0.05 was regarded as statistically significant. 

## 3. Results and Discussion

### 3.1. Characterization of Spray-dried Chitosan Cyclodextrins (SD-CS/CD) Composites

#### 3.1.1. Turbidity Measurements

The interaction between SD-CS and SBE-β-CD or HP-β-CD was studied based on changes in the turbidity in solutions, as the result of the precipitation of the inter-polymer complex (IPC) (Figure 1). The SD-CS solutions and SBE-β-CD solutions were transparent regardless of their concentration prior to the mixing. The transmittance of the mixing solutions showed no significant change with increasing SBE-β-CD concentration up to a SD-CS:SBE-β-CD ratio = 1:1. At higher ratios, the solution became cloudy (Figure 1b). Further increases in the amount of SBE-β-CD resulted in a decrease in absorbance. These results indicate that SD-CS formed a less-soluble IPC with SBE-β-CD predominantly at a 1:1 weight ratio, with the precipitation of micro-particles being observed. Thus, these results suggest that the formation of IPC could involve electrostatic interactions between the SO^3−^ group of SBE-β-CD and the NH^3+^ group of SD-CS. Actually, SD-CS contains about 170 monomer units of glucosamine per molecule, because the MW of SD-CS employed in this study was about 30000. Thus, the interaction between SD-CS and SBE-β-CD is maximal at a weight ratio of 1/1. These results suggest that about 13-14 glucosamine units per CS polymer participate in interactions with a SBE-β-CD molecule. On the other words, a sulfobutyl anion of SBE-β-CD interacted with about two cationic amino groups. Therefore, the SD-CS/SBE-β-CD composite was prepared by simply mixing both components at a ratio of 1/1, and this formulation was employed in subsequent studies. Otherwise, the relative absorption was essentially the same as the result of mixing the SD-CS and HP-β-CD solutions. 

#### 3.1.2. FT-IR Measurements

The IR spectrum of the SD-CS/SBE-β-CD composite showed that the peak at 1595 cm^−1^ assigned to the amine band of SD-CS was shifted to 1640 cm^−1^, indicating that the amine group was in the protonated form in IPC [22]. In addition, the FT-IR spectra of the SD-CS/SBE-β-CD composites were different from other spectra, with new peaks appearing at 1729 cm^−1^ and 1412 cm^−1^, consistent with CS/SBE-β-CD nano-particle studies (Figure 2) [23], indicating the formation of the SD-CS/ SBE-β-CD IPC. This IPC formation was further confirmed by differential scanning calorimetric studies, i.e., the endothermic peak of dehydration decreased from 140 °C–150 °C for CS and SBE-β-CD alone to 80 °C–120 °C for IPC, because of large hydrophobicity of the IPC surface due to the neutralization of the opposite charges of the components, facilitating the dehydration, as reported previously [15]. 

#### 3.1.3. Solubility Studies

To estimate the ability of SBE-β-CD or HP-β-CD to solubilize OLM, we investigated the interaction of the drug with SBE-β-CD and HP-β-CD in aqueous solution by the solubility method [10]. Figure 3 shows the phase solubility diagrams obtained for OLM/HP-β-, or /SBE-β-CD systems in water at 25 °C. The solubility plots showed an A_L_-type [24,25]. Thus, the solubility of OLM increased linearly with increasing concentration of the CD under these experimental methods. The apparent 1:1 stability constant (Kc) of the complexes was calculated from the initial linear portion of the solubility diagrams and the results indicated that SBE-β-CD (Kc = 128 M^−1^) had a higher solubilizing ability than HP-β-CD (Kc = 79 M^−1^). In fact, SBE-β-CD is a negatively charged derivative of β-CD, with an extended hydrophobic cavity and an extremely hydrophilic exterior surface in comparison to β-CD. This is because the secondary hydroxyls on the wide rim of β-CD are substituted with SBE groups. Furthermore, SBE-β-CD, with the advancements of higher aqueous solubility and the fact that it is non-toxic, is one of the most popular β-CD derivatives used as a pharmaceutical excipient [26,27]. Therefore, SBE-β-CD would be useful for not only as a solubilizer but also as an effective excipient.

### 3.2. In Vitro Release of OLM

A physical mixture of OLM, SD-CS, and SBE-β-CD or HP-β-CD in a weight ratio of 1:9:9 was compressed to form ordinary tablets. Figure 4 shows the dissolution profiles for OLM from the tablets in a pH 1.2 HCl solution, pH 6.8 phosphate buffer, and the sequential exposure to both solutions. As shown in Figure 4a, 100% of the OLM was released from the SD-CS/lactose or CS/HP-β-CD tablet after 2 h at pH 1.2, whereas less than 30% was released from the SD-CS/SBE-β-CD tablet. These phenomena were observed for physical mixtures of OLM, SD-CS and SBE-β-CD or HP-β-CD as well as for the tablets that were prepared, although the OLM in these physical mixtures were released slightly faster than from the tablets (data not shown). Furthermore, the release of OLM from a suspension of OLM/SBE-β-CD or OLM/HP-β-CD suddenly reached 100 % of the OLM release within a few minutes (data not shown). Similarly, the release of OLM from the SD-CS/SBE-β-CD tablet was significantly retarded in the case of the sequential exposure to both pH solutions (Figure 4c). In the case of the pH 1.2 HCl solution, the SD-CS/lactose or SD-CS/HP-β-CD tablet disintegrated rapidly and disappeared after 2 h, because SD-CS is quite soluble in acidic solutions. However, the SD-CS/SBE-β-CD tablet maintained its round shape in the pH 1.2 HCl solution after 2 h, although some erosion and disintegration was observed on the surface (data not shown). The erosion and disintegration gradually progressed and the tablet eventually became fully dispersed after about 10 h. These results suggest that, after exposure to water, the soluble SBE-β-CD and SD-CS in the solution at pH 1.2, easily formed a less-soluble IPC on the surface or inside the tablet, from which OLM was slowly dissolved as the erosion and disintegration of the tablet proceeded. Otherwise, the release of OLM from other type of tablet at pH 6.8 was significantly slower than that at pH 1.2, because SD-CS is sparingly soluble in neutral conditions (Figure 4a,b). The release from the SD-CS/lactose or SD-CS/HP-β-CD tablet at pH 6.8 was accompanied by the erosion and disintegration of the tablet, and the surface of the tablet was encapsulated in a thin gel. On the other hand, the round shape of the SD-CS/SBE-β-CD tablet did not undergo disintegration and the shape was maintained for the duration of the experiment (Figure 4d). In general, CS is known to have gelation properties, although not high. In a previous study, we reported that chitin nanofibers formed strong elastic gels with SBE-β-CD [28,29]. Therefore, these results indicate that the gelation that was observed on the surface of SD-CS/SBE-β-CD tablets after exposure to water, which functioned as a barrier to water penetration, caused the drug release to decelerate. 

### 3.3. In Vivo Release of Olmesartan Tablets (OLM)

The concentration of OLM in plasma following the administration of suspensions of OLM, OLM/SD-CS/lactose, OLM/SD-CS/HP-β-CD, and OLM/SD-CS/SBE-β-CD are shown in Figure 5 and the pharmacokinetic parameters, as determined from OLM plasma concentration-time data, are presented in Table 1. The results indicate that the ratio of the AUC_(0–∞)_ for the OLM/CS/SBE-β-CD to the AUC for the drug alone or the OLM/SD-CS/lactose was accompanied by the increase in C_max_ values. Furthermore, a significant difference in T_max_, AUC_(0–∞)_, and half-life (T_1/2_) was observed between the OLM/SD-CS/HP-β-CD and OLM/SD-CS/SBE-β-CD. The insignificant difference in plasma concentrations of OLM was observed between these formulations until about 3 h after the administration. This may be due to the fact that the release rate of the drug from the SD-CS/ SBE-β-CD composite is not markedly slowed down in pH 1.2, when compared with that in pH 6.8, as shown in Figure 4c. On the other hand, the markedly slowed release of the drug from the composite in the neutral solution (pH6.8) after 3 h might contribute significantly to the extended plasma concentrations of OLM following the administration of the composite.

### 3.4. In Vivo Antihypertensive Effects of OLM

The *in vivo* antihypertensive effect of OLM was evaluated using hypertensive model rats, and the results are shown in Figure 6. In hypertensive rats, the mean blood pressure (MBP) for the OLM alone, OLM/SD-CS/lactose, and olmesartan/SD-CS/HP-β-CD systems returned to the control level within 24 h, whereas in the OLM/SD-CS/SBE-β-CD-treated groups, a large continuous decrease in this parameter was noted, compared with the control and other composites. This effect can be attributed to the increased solubility of OLM and the extended-release effect, as the result of inter-polymer complex formation between SD-CS and SBE-β-CD (Figure 6, Table 1). Because CDs are known to function as a potent absorption enhancer [19], these phenomena might be another reasonable explanation for the increment in the oral bioavailability of OLM in SBE-β-CD. Actually, the solubility of OLM was significantly enhanced by SBE-β-CD, and was linearly dependent on the CD concentration (A_L_ type) [20]. In fact, we also indicated that the SBE-β-CD (128 M^−1^) had a higher solubilizing ability than that of HP-β-CD (79 M^−1^) (Figure 3). Therefore, these results suggest that a simple mixing of SD-CS and SBE-β-CD can be potentially useful for the controlled release of a medicine for the continuous treatment of hypertension.

## 4. Conclusions

To design a more effective and safer process for the oral administration of OLM, we prepared a sustained-release tablet of OLM by the direct compression of a SD-CS/SBE-β-CD composite and evaluated its anti-hypertensive effect. The release of OLM from tablets of the SD-CS/SBE-β-CD composite was slow in media at both pH 1.2 and 6.8. The *in vitro* slow release characteristics of the SD-CS/SBE-β-CD composite were reflected in the *in vivo* absorption of the drug after oral administration to rats. Furthermore, the SD-CS/SBE-β-CD composite continuously increased the antihypertensive effect of OLM, compared with that of the OLM itself. Considering the multiple functions of SD-CS and SBE-β-CD, a simple mixing of SD-CS and SBE-β-CD represents a potentially useful process for creating an effective and safe excipient of OLM for the continuous treatment of hypertension.

## Figures and Tables

**Figure 1 pharmaceutics-11-00082-f001:**
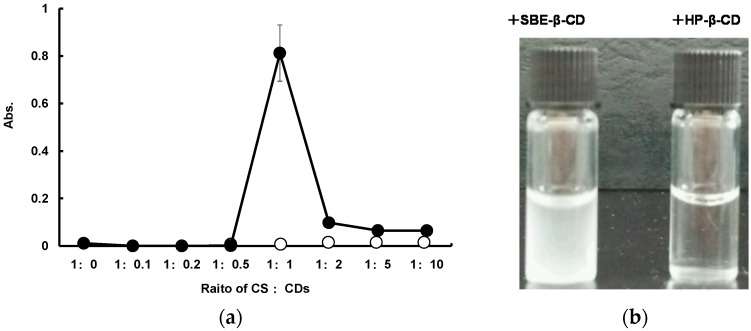
Effect of the weight ratio of spray-dried chitosan (SD-CS) and sulfobutyl ether-β-cyclodextrin (SBE-β-CD) (close circles) or HP-β-CD (open circles) on the absorbance as a measurements of turbidity of the solution (**a**) and appearance of SD-CS/SBE-β-CD solution (left) and SD-CS/HP-β-CD solution (right) (weight ratio = 1:1) (**b**).

**Figure 2 pharmaceutics-11-00082-f002:**
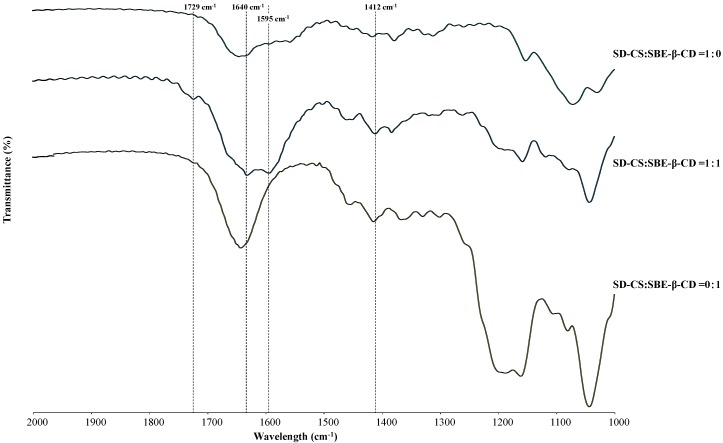
Fourier transform infrared spectroscopy (FT-IR) spectra of spray-dried chitosan (SD-CS), sulfobutyl ether-β-cyclodextrin (SBE-β-CD), and SD-CS/SBE-β-CD components.

**Figure 3 pharmaceutics-11-00082-f003:**
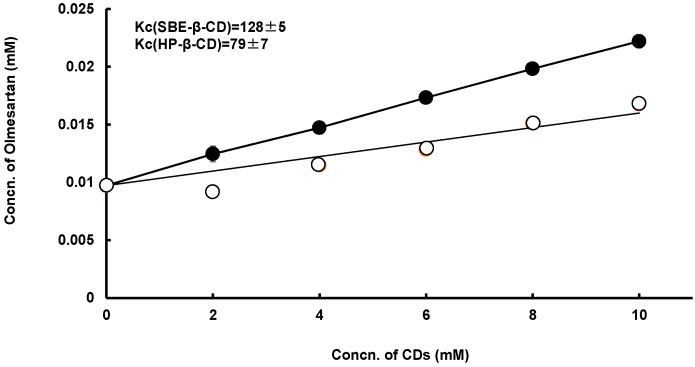
Phase solubility diagrams of olmesartan tablets (OLM)-sulfobutyl ether-β-cyclodextrin (SBE-β-CD), OLM/ SBE-β-CD (close circle) and OLM/HP-β-CD (open circle) systems in water at 25 °C.

**Figure 4 pharmaceutics-11-00082-f004:**
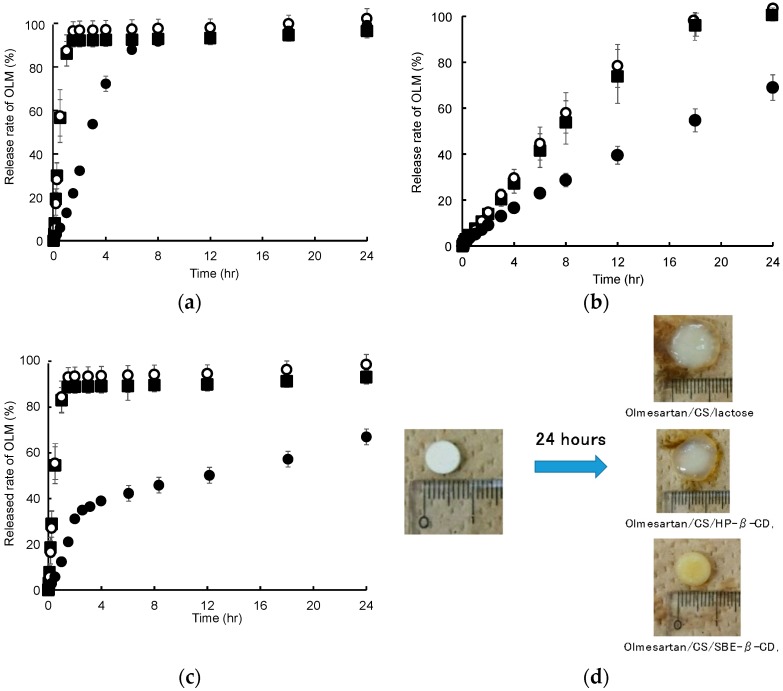
Release profiles for olmesartan tablets (OLM) from tablets at pH 1.2 (**a**), pH 6.8 (**b**), sequential exposure to pH 1.2 and pH 6.8 (**c**) and microscopic observation of OLM/SD-CS/lactose, OLM/SD-CS/HP-β-CD, and OLM/SD-CS/SBE-β-CD tablets before (left) and after 24 h (right) in media at pH 6.8 (**d**). OLM/ SD-CS/SBE-β-CD (close circle), OLM/SD-CS/HP-β-CD (open circle), and OLM/SD-CS/lactose (close square). (SD-CS = spray-dried chitosan, SBE-β-CD = sulfobutyl ether-β-cyclodextrin).

**Figure 5 pharmaceutics-11-00082-f005:**
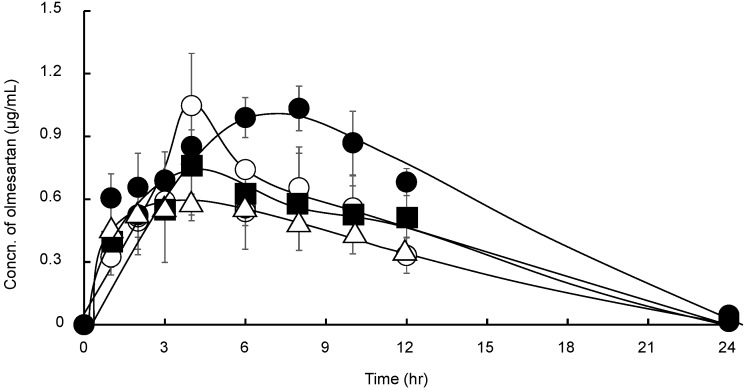
Mean plasma pharmacokinetic profiles of OLM (△), OLM/SD-CS/lactose (■), OLM/SD-CS/HP-β-CD (○), OLM/SD-CS/SBE-β-CD (●) suspensions in rats (at OLM dose of 8 mg/kg). (OLM = olmesartan tablets, SBE-β-CD = sulfobutyl ether-β-cyclodextrin, and SD-CS = spray-dried chitosan).

**Figure 6 pharmaceutics-11-00082-f006:**
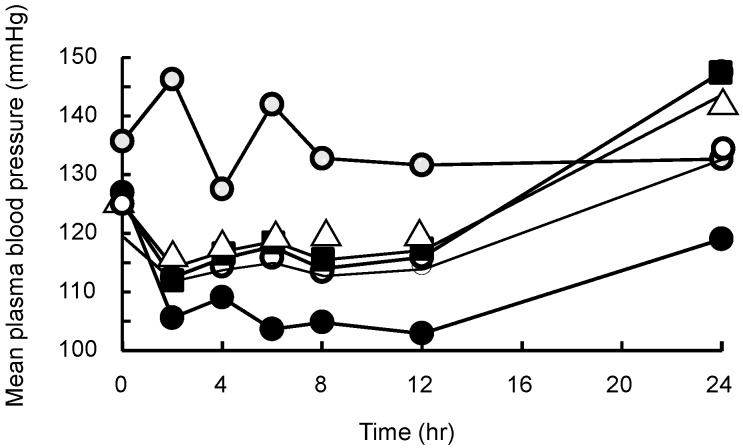
Mean plasma blood pressure profiles for control (●), OLM suspension (△), OLM/SD-CS/lactose (■), OLM/SD-CS/HP-β-CD (○), OLM/SD-CS/SBE-β-CD (●) suspensions in rats (at OLM dose of 1 mg/kg). (OLM = olmesartan tablets, SBE-β-CD = sulfobutyl ether-β-cyclodextrin, and SD-CS = spray-dried chitosan).

**Table 1 pharmaceutics-11-00082-t001:** Pharmacokinetic parameters for olmesartan tablets (OLM) in rats following oral administration of the suspensions (8 mg/kg as the drug).

	AUC_(0-∞)_ (mgh/mL)	C_max_ (μg/mL)	T_1/2_ (h)	T_max_ (h)
OLM suspension	7.8 ± 0.10	0.80 ± 0.16	3.6 ± 0.11	3.3 ± 0.16
OLM/SD-CS/lactose	8.2 ± 0.80	0.74 ± 0.12	4.5 ± 0.34	3.6 ± 0.17
OLM/SD-CS/HP-β-CD	9.4 ± 0.58 ^b^	1.0 ± 0.033 ^a^	3.0 ± 0.45	3.3 ± 0.26
OLM/SD-CS/SBE-β-CD	14.1 ± 1.2 ^a,d,e^	1.0 ± 0.10 ^b,d^	4.2 ± 0.29 ^b,e^	5.0 ± 0.10 ^b,c,e^

^a^*p* < 0.01 for OLM suspension, ^b^
*p* < 0.05 for OLM suspension, ^c^
*p* < 0.01 for OLM /SD-CS/lactose, ^d^
*p* < 0.05 for OLM /SD-CS/lactose, ^e^
*p* < 0.05 for OLM /SD-CS/HP-β-CD.

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
