# Peer review of "Design and Evaluation of An Extended-Release Olmesartan Tablet Using Chitosan/Cyclodextrin Composites"

_pharmaceutics, 2019, doi:10.3390/pharmaceutics11020082_

Round 1

Reviewer 1 Report

The authors have designed sustained-realise tablets of Olmesartan obtained by the direct compression of composites  of  anionic  sulfobutyl  ether-β-cyclodextrin  (SBE-β-CD)  and  cationic  chitosan (CS).

 The research objectives are good however there are number of lacunae and queries which need to be addressed. Also the manuscript has too many gaps in the draft and needs to be completely rewritten with greater scientific rigor.

Introduction should be improved because it is not a fluent reading and it does not provide sufficient background. Moreover In the whole article there are many typing errors to fix.

The work is lacking in studies on physicochemical characterization of inter-polymer complex (IPC) obtained, so that justified the sentences on pg 3 lines 127-129,  “These  results suggest that the formation of IPC could involve electrostatic interactions between the SO 3- group of SBE-β-CD and the NH 3+  group of CS”  (e.g. FT-IR, light scattering, thermal analysis).

Moreover, it is not clear in the whole manuscript whether the Authors first form the of inter-polymer complex (IPC) and then they combine the drug to make the tablet, or the tablet is formed by a simple physical mixture of all the components. In analogy, how are prepared the suspensions given to rats? In fact, is unclear the means of sentences pg 3 lines 134-137.

Some details in methods should be also added.

2.4 paragraph: Why only CS were spray-dried ?

The tablet types described in this paragraph do not match those discussed in the results. In this paragraph it seems that more types have been prepared.

2.6 paragraph: the project identification code, date of approval and name of the ethics committee or institutional review board for animal experiments is missing.

Is missed also the experimental design with the group of animals and relatively dosage form administered.

A briefly description of HPLC method should be reported (with the modifications that they affirm to have done).

2.7 paragraph: number of animals? is there a control group?

the abscissa of the graph in Fig1 shows the absorbance while the legend reports "transmittance".

However some improvement is needed.

Author Response

Reply to Reviewer #1:

(1)   In response to the comment, “Introduction should be improved because it is not a fluent reading and it does not provide sufficient background. Moreover In the whole article there are many typing errors to fix.”, we added additional text in the Introduction as follows line 33-34 and line 43-48. Typing and grammar errors were also corrected by an editing service (Dr. Milton S Feather, Technical Editing Services).

(2)   In response to the comment, “The work is lacking in studies on physicochemical characterization of inter-polymer complex (IPC) obtained, so that justified the sentences on pg 3 lines 127-129,  “These results suggest that the formation of IPC could involve electrostatic interactions between the SO3- group of SBE-β-CD and the NH3+ group of CS”  (e.g. FT-IR, light scattering, thermal analysis)”, we added an additional Figure and text in the Materials, Results and Discussion as follows line 85-88, and line 163-181.

(3)   In response to the comment, “Moreover, it is not clear in the whole manuscript whether the Authors first the form of inter-polymer complex (IPC) and then they combine the drug to make the tablet, or the tablet is formed by a simple physical mixture of all the components. In analogy, how are prepared the suspensions given to rats? In fact, is unclear the means of sentences pg 3 lines 134-137.”, we added additional text in the Materials as follows line 91-98 and line 110-111.

(4)   In response to the comment, “2.4 paragraph: Why only CS were spray-dried ? The tablet types described in this paragraph do not match those discussed in the results. In this paragraph it seems that more types have been prepared.”, we added additional explanatory text on lines 65-66 and changed the word from CS to CS, because we used spray-dried chitosan (SD-CS) in all experiments.

(5)   In response to the comment, “2.6 paragraph: the project identification code, date of approval and name of the ethics committee or institutional review board for animal experiments is missing.”, we added additional text on lines122-124.

(6)   In response to the comment, “Is missed also the experimental design with the group of animals and relatively dosage form administered. A briefly description of HPLC method should be reported (with the modifications that they affirm to have done).”, we added additional text as follows on lines 113-117.

(7)   In response to the comment, “2.7 paragraph: number of animals? is there a control group?”, we added additional text in the material and methods on lines 127-129.

(8)   In response to the comment, “the abscissa of the graph in Fig1 shows the absorbance while the legend reports "transmittance"..”, we corrected the caption for Fig.1 (line 160).

Reviewer 2 Report

This manuscript describes the preparation, in vitro dissolution and in vivo absorption studies of CS-SBE-β-CS formulations for extended drug-release. The publication of these results are recommended after the following revisions:

L90: It is unclear whether the sampling was automated or not. Was the volume of the dissolution media kept constantly at 450 mL during the tests or it decreased with every samples taken?

L98: The part “Materials and methods” usually serve to inform the reader about the methods and materials used in the examined work. By comparison, quite a lot of abbreviation are found in this part of the work referring to other publications which takes a lot more time to understand the methods. Please instead of referring to other works complete the “Materials and methods” part with the exact descriptions of the following methods:

·       L71: solubility measurements referring to [15]

·       L81: spray drying of CS solutions referring to [17]

·       L98: samples prepared for HPLC analysis referring to [18]

L126-127: What was the absorbance of the transparent CS and SBE-β-CD solutions? What do the authors mean by “with micro-particles being precipitated”? If precipitation occurred with the sample containing the IPC components at a weight ratio of 1:1 then maybe 1:1 is not the most favorable composition since at least one of the components precipitated. Please explain this issue.

L134: It is described in the introductory part that the filtration of nanoparticles is cumbersome using filter paper. The authors tried to conduct the same method with microparticles unsuccessfully. Is there any other way to isolate the prepared microparticles?

L166: It is not clear how the sequential exposure of the tablets to different pH solutions was conducted. Was the tablets first put into pH=1.2 dissolution media and then taken out and put into pH=6.8 dissolution media or was the pH of the pH=1.2 dissolution media increased to pH=6.8?

L167: In figure 3a it is clearly observable that the dissolution of OLM does not reach 100% as stated but 85-90%. However, in figure 3b two of the compositions show 100% drug release after 24 hours. What is the explanation for this phenomenon? The legend is missing in figure 3. Please correct.

L194: Since the in vivo release of OLM was investigated using suspensions of the physical mixtures of OLM, CS and CD’s it would have been interesting to conduct the in vitro dissolution tests with these formulations besides tablets. The reason for this is that the rapidly formed less-soluble CS-SBE-β-CD inter-polymer complex described starting from L176 might not be as effective for extended release in case of a loose suspension formulation as in case of a tight tablet.

It would have been also interesting to investigate the dissolution and the absorption profile of HP-β-CD-OLM and SBE-β-CD-OLM formulations without the introduction of CS since CDs alone can enhance dissolution and absorption properties of drugs as described.

Author Response

Reply to Reviewer #2:

(1)   In response to the comment, “L90: It is unclear whether the sampling was automated or not. Was the volume of the dissolution media kept constantly at 450 mL during the tests or it decreased with every samples taken?”, we added additional text in the Materials and Methods as follows line 104-106.

(2)   In response to the comment, “The part “Materials and methods” usually serve to inform the reader about the methods and materials used in the examined work. By comparison, quite a lot of abbreviation are found in this part of the work referring to other publications which takes a lot more time to understand the methods. Please instead of referring to other works complete the “Materials and methods” part with the exact descriptions of the following methods: L71: solubility measurements referring to [15], L81: spray drying of CS solutions referring to [17], L98: samples prepared for HPLC analysis referring to [18]”, we added additional text in the “Materials and methods” as follows on lines 76-81, lines 91-98, and lines 113-117.

(3)   In response to the comment, “L126-127: What was the absorbance of the transparent CS and SBE-β-CD solutions? What do the authors mean by “with micro-particles being precipitated”? If precipitation occurred with the sample containing the IPC components at a weight ratio of 1:1 then maybe 1:1 is not the most favorable composition since at least one of the components precipitated. Please explain this issue.”, we corrected these sentences and FT-IR data have been added in the results and discussion as follows on lines 85-88, lines 147-150, lines 157-158, and lines 163-181.

(4)   In response to the comment, “L134: It is described in the introductory part that the filtration of nanoparticles is cumbersome using filter paper. The authors tried to conduct the same method with microparticles unsuccessfully. Is there any other way to isolate the prepared microparticles?”, we deleted these sentences because this text and description is inadequate.

(5)   In response to the comment, “L166: It is not clear how the sequential exposure of the tablets to different pH solutions was conducted. Was the tablets first put into pH=1.2 dissolution media and then taken out and put into pH=6.8 dissolution media or was the pH of the pH=1.2 dissolution media increased to pH=6.8?”, we added additional text to the caption in Figure 4 (line 230-231).

(6)   In response to the comment, “L167: In figure 3a it is clearly observable that the dissolution of OLM does not reach 100% as stated but 85-90%. However, in figure 3b two of the compositions show 100% drug release after 24 hours. What is the explanation for this phenomenon? The legend is missing in figure 3. Please correct.”, we corrected the Y axis of Figure 4.

(7)   In response to the comment, “L194: Since the in vivo release of OLM was investigated using suspensions of the physical mixtures of OLM, CS and CD’s it would have been interesting to conduct the in vitro dissolution tests with these formulations besides tablets. The reason for this is that the rapidly formed less-soluble CS-SBE-β-CD inter-polymer complex described starting from L176 might not be as effective for extended release in case of a loose suspension formulation as in case of a tight tablet. It would have been also interesting to investigate the dissolution and the absorption profile of HP-β-CD-OLM and SBE-β-CD-OLM formulations without the introduction of CS since CDs alone can enhance dissolution and absorption properties of drugs as described.”: we added additional text in the Results and Discussion (line 205-209).  It should also be noted that we also feel that the comments made by the reviewer are important and will need to be investigated in the future, if this system is to be of commercial value.  Our current plans are to conduct further, in-depth investigations regarding this system that will involve, among other issues the experiments alluded to above.  Additional in vitro dissolution tests will need to be much more extensive and, we feel, would be beyond the scope of this current study.

Reviewer 3 Report

The authors designed sustained-release tablets of olmesartan with sulfobutyl ether-β-cyclodextrin (SBE-β-CD) and cationic and chitosan (CS). The whole study was designed well and presented with supportive data. However, there are few minor comments that authors need to address.

1 the drug content in the tablet should be tested.

2 the characterization on tablet should be supplemented with size, thickness and hardness.

3 the measurements of turbidity should be explained with more discussions to make it clear to reader. Like the difference between the SBE-β-CD or HP-β-CD.

4 the label above the figure 1b is not consistent with the caption below. Author should correct them.

5 the halt-life should be provided for PK data shown in table 1.

Author Response

Reply to Reviewer #3:

(1)   In response to the comment, “The drug content in the tablet should be tested. The characterization on tablet should be supplemented with size, thickness and hardness.”, we added additional text in the Materials and Methods as follows on lines 91-98.

(2)   In response to the comment, “The measurements of turbidity should be explained with more discussions to make it clear to reader. Like the difference between the SBE-β-CD or HP-β-CD.”, we added additional text in “results and discussion” as follows on lines 147-150.

(3)   In response to the comment, “The label above the figure 1b is not consistent with the caption below. Author should correct them.”, we added “b” in Figure 1 (line 161).

(4)   In response to the comment, “The halt-life should be provided for PK data shown in table 1.”, we added “Half-life” in Table 1.

Round 2

Reviewer 1 Report

This manuscript is well revised based on the reviewer's comment. However minor revisions must be made to accept the manuscript for publication.

In particular:

-        The manuscript describes two different types of animal experiments in paragraph 2.7 and 2.8. In paragraph 2.7. “In vivo pharmacokinetic studies of SD-CS/CDs /OLM composite” the experiments are carried out on an undefined type of rat and is missed the project identification code and name of the ethics committee. In paragraph 2.8 the same information is reported.  Authors should give more details in paragraph 2.7

-         Lines 211-214: Authors describe DSC analysis carried out in previous studies, but, if the data have not been published, they should report the details of the DSC analysis in a different paragraph of this manuscript and possibly show the diagrams.

Although the manuscript has been revised by an Editing Service, there are still some unclear or repetitive sentences. In particular:

Lines 82-83: “the level of OLM determined by determining the concentration by UV spectroscopy at 248 nm”, this sentence must be rewritten and UV is a spectrophotometry analysis. 

Lines 113-114.  a part of the sentence is repeated twice

Line: 123: the instrument brand has already been described previously and must be omitted here.

Lines 131: the type and brand of the column has been omitted

Line 133-134: the operations described are obvious, so they must be deleted.

Legend of Fig. 4: the last line (c) and superfluous.

Legend of Fig. 5: it is unclear. it should be:   “Mean plasma pharmacokinetic profiles of ……..   following oral administration of suspensions in rats (at OLM dose of 8 mg/kg)

Legend of Fig. 6: it is unclear. it should be:  “Mean plasma blood pressure profiles for… following oral administration of suspensions in rats (at OLM dose of 8 mg/kg)

Line 372:  “CD/SBE-b-CD” I think it's a typos.

Author Response

Reply to comments from Reviewers:

We are appreciative of the Reviewer’s helpful comments regarding our manuscriptDesign and evaluation of an extended-release olmesartan tablet using chitosan/cyclodextrin composites” (Manuscript ID: pharmaceutics-394965). We carefully considered their comments and prepared a revised version of our manuscript, prepared in the light of your comments. The following is a point-by-point list of responses to the comments. In the revised manuscript, the revised sections or sentences are underlined.

Reply to Reviewer #1:

(1)   In response to the comment, “The manuscript describes two different types of animal experiments in paragraph 2.7 and 2.8. In paragraph 2.7. “In vivo pharmacokinetic studies of SD-CS/CDs /OLM composite” the experiments are carried out on an undefined type of rat and is missed the project identification code and name of the ethics committee. In paragraph 2.8 the same information is reported.  Authors should give more details in paragraph 2.7”, we added additional text in the Materials and Methods as follows on lines 112-115.

(2)   In response to the comment, “Lines 211-214: Authors describe DSC analysis carried out in previous studies, but, if the data have not been published, they should report the details of the DSC analysis in a different paragraph of this manuscript and possibly show the diagrams.”, actually, we previously published DSC data, so, we added Ref. 15 in line 171 and 172.

(3)   In response to the comment, “Lines 82-83: “the level of OLM determined by determining the concentration by UV spectroscopy at 248 nm”, this sentence must be rewritten and UV is a spectrophotometry analysis.”, we corrected these sentences as follows on lines 83-84.

(4)   In response to the comment, “Lines 113-114.  a part of the sentence is repeated twice”,”Line: 123: the instrument brand has already been described previously and must be omitted here.”, “Lines 131: the type and brand of the column has been omitted”, “Line 133-134: the operations described are obvious, so they must be deleted.”, “Legend of Fig. 4: the last line (c) and superfluous.”, we deleted these sentences because this text and description is inadequate.

(5)   In response to the comment, “Legend of Fig. 5 and Fig. 6: it is unclear. it should be: “Mean plasma pharmacokinetic profiles of ……..   following oral administration of suspensions in rats (at OLM dose of 8 mg/kg) and (at OLM dose of 8 mg/kg)”, we corrected the caption in Figure 5 and 6 (line 253 and 275-6).

(6)   In response to the comment, “Line 372: “CD/SBE-b-CD” I think it's a typos.”, we corrected the line 280.

(7)   Concerning the quality of the English used in the paper, we had the paper reviewed by an outside editing service and a letter of confirmation from them is attached.
